# Understanding Attention and Generalization in Graph Neural Networks

**Boris Knyazev**
University of Guelph
Vector Institute
bknyazev@uoguelph.ca

**Graham W. Taylor**
University of Guelph
Vector Institute, Canada CIFAR AI Chair
gwtaylor@uoguelph.ca

**Mohamed R. Amer**[*]
Robust.AI
mohamed@robust.ai

## Abstract

We aim to better understand attention over nodes in graph neural networks (GNNs) and identify factors influencing its effectiveness. We particularly focus on the ability of attention GNNs to generalize to larger, more complex or noisy graphs. Motivated by insights from the work on Graph Isomorphism Networks, we design simple graph reasoning tasks that allow us to study attention in a controlled environment. We find that under typical conditions the effect of attention is negligible or even harmful, but under certain conditions it provides an exceptional gain in performance of more than 60% in some of our classification tasks. Satisfying these conditions in practice is challenging and often requires optimal initialization or supervised training of attention. We propose an alternative recipe and train attention in a weakly-supervised fashion that approaches the performance of supervised models, and, compared to unsupervised models, improves results on several synthetic as well as real datasets. Source code and datasets are available at `https://github.com/bknyaz/graph_attention_pool`.

## 1 Attention meets pooling in graph neural networks

The practical importance of attention in deep learning is well-established and there are many arguments in its favor [1], including interpretability [2, 3]. In graph neural networks (GNNs), attention can be defined over edges [4, 5] or over nodes [6]. In this work, we focus on the latter, because, despite being equally important in certain tasks, it is not as thoroughly studied [7]. To begin our description, we first establish a connection between attention and pooling methods. In convolutional neural networks (CNNs), pooling methods are generally based on uniformly dividing the regular grid (such as one-dimensional temporal grid in audio) into local regions and taking a single value from that region (average, weighted average, max, stochastic, etc.), while attention in CNNs is typically a separate mechanism that weights $C$-dimensional input $X \in \mathbb{R}^{N \times C}$:

$$Z = \alpha \odot X, \tag{1}$$

where $Z_i = \alpha_i X_i$ - output for unit (node in a graph) $i$, $\sum_i^N \alpha_i = 1$, $\odot$ - element-wise multiplication, $N$ - the number of units in the input (i.e. number of nodes in a graph).

In GNNs, pooling methods generally follow the same pattern as in CNNs, but the pooling regions (sets of nodes) are often found based on clustering [8, 9, 10], since there is no grid that can be uniformly divided into regions in the same way across all examples (graphs) in the dataset. Recently, top-k pooling [11] was proposed, diverging from other methods: instead of clustering "similar" nodes, it propagates only part of the input and this part is not uniformly sampled from the input. Top-k pooling can thus select some local part of the input graph, completely ignoring the rest. For this reason at first glance it does not appear to be logical.

---

[*]Most of this work was done while the author was at SRI International.

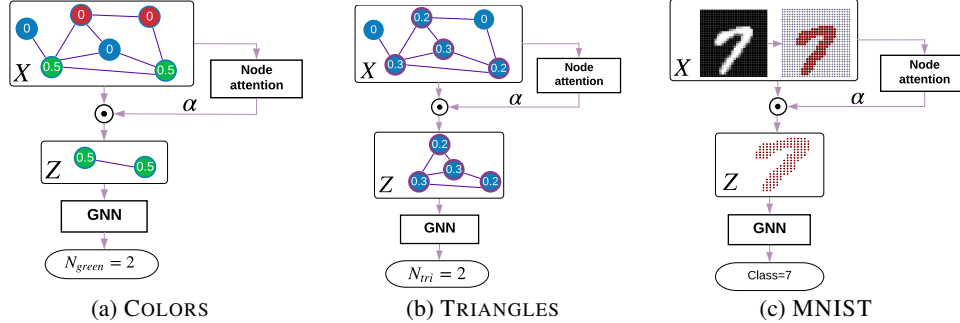

(a) COLORS      (b) TRIANGLES      (c) MNIST

Figure 1: Three tasks with a controlled environment we consider in this work. The values inside the nodes are ground truth attention coefficients, $\alpha_i^{GT}$, which we find heuristically (see Section 3.1).

However, we can notice that pooled feature maps in [11, Eq. 2] are computed in the same way as attention outputs $Z$ in Eq. 1 above, if we rewrite their Eq. 2 in the following way:

$$Z_i = \begin{cases} \alpha_i X_i, & \forall i \in P \\ \emptyset, & \text{otherwise,} \end{cases} \tag{2}$$

where $P$ is a set of indices of pooled nodes, $|P| \le N$, and $\emptyset$ denotes the unit is absent in the output.

The only difference between Eq. 2 and Eq. 1 is that $Z \in \mathbb{R}^{|P| \times C}$, i.e. the number of units in the output is smaller or, formally, there exists a ratio $r = |P|/N \le 1$ of preserved nodes. We leverage this finding to integrate attention and pooling into a unified computational block of a GNN. In contrast, in CNNs, it is challenging to achieve this, because the input is defined on a regular grid, so we need to maintain resolution for all examples in the dataset after each pooling layer. In GNNs, we can remove any number of nodes, so that the next layer will receive a smaller graph. When applied to the input layer, this form of attention-based pooling also brings us interpretability of predictions, since the network makes a decision only based on pooled nodes.

Despite the appealing nature of attention, it is often unstable to train and the conditions under which it fails or succeeds are unclear. Motivated by insights of [12] recently proposed Graph Isomorphism Networks (GIN), we design two simple graph reasoning tasks that allow us to study attention in a controlled environment where we know ground truth attention. The first task is counting colors in a graph (COLORS), where a color is a unique discrete feature. The second task is counting the number of triangles in a graph (TRIANGLES). We confirm our observations on a standard benchmark, MNIST [13] (Figure 1), and identify factors influencing the effectiveness of attention.

Our synthetic experiments also allow us to study the ability of attention GNNs to generalize to larger, more complex or noisy graphs. Aiming to provide a recipe to train more effective, stable and robust attention GNNs, we propose a weakly-supervised scheme to train attention, that does not require ground truth attention scores, and as such is agnostic to a dataset and the choice of a model. We validate the effectiveness of this scheme on our synthetic datasets, as well as on MNIST and on real graph classification benchmarks in which ground truth attention is unavailable and hard to define, namely COLLAB [14, 15], PROTEINS [16], and D&D [17].

## 2 Model

We study two variants of GNNs: Graph Convolutional Networks (GCN) [18] and Graph Isomorphism Networks (GIN) [12]. One of the main ideas of GIN is to replace the MEAN aggregator over nodes, such as the one in GCN, with a SUM aggregator, and add more fully-connected layers after aggregating neigboring node features. The resulting model can distinguish a wider range of graph structures than previous models [12, Figure 3].

### 2.1 Thresholding by attention coefficients

To pool the nodes in a graph using the method from[11] a predefined ratio $r = |P|/N$ (Eq. 2) must be chosen for the entire dataset. For instance, for $r = 0.8$ only 80% of nodes are left after each pooling

layer. Intuitively, it is clear that this ratio should be different for small and large graphs. Therefore, we propose to choose threshold $\tilde{\alpha}$, such that only nodes with attention values $\alpha_i > \tilde{\alpha}$ are propagated:

$$Z_i = \begin{cases} \alpha_i X_i, & \forall i : \alpha_i > \tilde{\alpha} \\ \emptyset, & \text{otherwise.} \end{cases} \tag{3}$$

Note, that dropping nodes from a graph is different from keeping nodes with very small, or even zero, feature values, because a bias is added to node features after the following graph convolution layer affecting features of neighbors. An important potential issue of dropping nodes is the change of graph structure and emergence of isolated nodes. However, in our experiments we typically observe that the model predicts similar $\alpha$ for nearby nodes, so that an entire local neighborhood is pooled or dropped, as opposed to clustering-based methods which collapse each neighborhood to a single node. We provide a quantitative and qualitative comparison in Section 3.

## 2.2 Attention subnetwork

To train an attention model that predicts the coefficients for nodes, we consider two approaches: (1) Linear Projection [11], where a single layer projection $\mathbf{p} \in \mathbb{R}^C$ is trained: $\alpha_{pre} = X\mathbf{p}$; and (2) DiffPool [10], where a separate GNN is trained:

$$\alpha_{pre} = \text{GNN}(A, X), \tag{4}$$

where $A$ is the adjacency matrix of a graph. In all cases, we use a softmax activation [1, 2] instead of tanh in [11], because it provides more interpretable results and ecourages sparse outputs: $\alpha = \text{softmax}(\alpha_{pre})$. To train attention in a supervised or weakly-supervised way, we use the Kullback-Leibler divergence loss (see Section 3.3).

## 2.3 ChebyGIN

In some of our experiments, the performance of both GCNs and GINs is quite poor and, consequently, it is also hard for the attention subnetwork to learn. By combining GIN with ChebyNet [8], we propose a stronger model, ChebyGIN. ChebyNet is a multiscale extension of GCN [18], so that for the first scale, $K = 1$, node features are node features themselves, for $K = 2$ features are averaged over one-hop neighbors, for $K = 3$ - over two-hop neighbors and so forth. To implement the SUM aggregator in ChebyGIN, we multiply features by node degrees $D_i = \sum_j A_{ij}$ starting from $K = 2$. We also add more fully-connected layers after feature aggregation as in GIN.

# 3 Experiments

We introduce the color counting task (COLORS) and the triangle counting task (TRIANGLES) in which we generate synthetic training and test graphs. We also experiment with MNIST images [13] and three molecule and social datasets. In COLORS, TRIANGLES and MNIST tasks (Figure 1), we assume to know ground truth attention, i.e. for each node $i$ we heuristically define its importance in solving the task correctly, $\alpha_i^{GT} \in [0, 1]$, which is necessary to train (in the supervised case) and evaluate our attention models.

## 3.1 Datasets

**COLORS.** We introduce the color counting task. We generate random graphs where features for each node are assigned to one of the three one-hot values (colors): [1,0,0] (red), [0,1,0] (green), [0,0,1] (blue). The task is to count the number of green nodes, $N_{green}$. This is a trivial task, but it lets us study the influence of initialization of the attention model $\mathbf{p} \in \mathbb{R}^3$ on the training dynamics. In this task, graph structure is unimportant and edges of graphs act like a medium to exchange node features. Ground truth attention is $\alpha_i^{GT} = 1/N_{green}$, when $i$ corresponds to green nodes and $\alpha_i^{GT} = 0$ otherwise. We also extend this dataset to higher $n$-dimensional cases $\mathbf{p} \in \mathbb{R}^n$ to study how model performance changes with $n$. In these cases, node features are still one-hot vectors and we classify the number of nodes where the second feature is one.

**TRIANGLES.** Counting the number of triangles in a graph is a well-known task which can be solved analytically by computing $\text{trace}(A^3)/6$, where $A$ is an adjacency matrix. This task turned out to

be hard for GNNs, so we add node degree features as one-hot vectors to all graphs, so that the model can exploit both graph structure and features. Compared to the COLORS task, here it is more challenging to study the effect of initializing $\mathbf{p}$, but we can still calculate ground truth attention as $\alpha_i^{GT} = T_i / \sum_i T_i$, where $T_i$ is the number of triangles that include node $i$, so that $\alpha_i^{GT} = 0$ for nodes that are not part of triangles.

**MNIST-75SP.** MNIST [13] contains 70k grayscale images of size $28 \times 28$ pixels. While each of 784 pixels can be represented as a node, we follow [19, 20] and consider an alternative approach to highlight the ability of GNNs to work on irregular grids. In particular, each image can be represented as a small set of superpixels without losing essential class-specific information (see Figure 2). We compute SLIC [21] superpixels for each image and build a graph, in which each node corresponds to a superpixel with node features being pixel intensity values and coordinates of their centers of masses. We extract $N \leq 75$ superpixels, hence the dataset is denoted as MNIST-75SP. Edges are formed based on spatial distance between superpixel centers as in [8, Eq. 8]. Each image depicts a handwritten digit from 0 to 9 and the task is to classify the image. Ground truth attention is considered to be $\alpha_i^{GT} = 1 / N_{nonzero}$ for superpixels with nonzero intensity, and $N_{nonzero}$ is the total number of such superpixels. The idea is that only nonzero superpixels determine the digit class.

**Molecule and social datasets.** We extend our study to more practical cases, where ground truth attention is not available, and experiment with protein datasets: PROTEINS [16] and D&D [17], and a scientific collaboration dataset, COLLAB [14, 15]. These are standard graph classification benchmarks. A standard way to evaluate models on these datasets is to perform 10-fold cross-validation and report average accuracy [22, 10]. In this work, we are concerned about a model's ability to generalize to larger and more complex or noisy graphs, therefore, we generate splits based on the number of nodes. For instance, for PROTEINS we train on graphs with $N \leq 25$ nodes and test on graphs with $6 \leq N \leq 620$ nodes (see Table 2 for details about splits of other datasets and results).

A detailed description of tasks and model hyperparameters is provided in the *Supp. Material*.

## 3.2 Generalization to larger and noisy graphs

One of the core strengths of attention is that it makes it easier to generalize to unseen, potentially more complex and/or noisy, inputs by reducing them to better resemble certain inputs in the training set. To examine this phenomenon, for COLORS and TRIANGLES tasks we add test graphs that can be several times larger (TEST-LARGE) than the training ones. For COLORS we further extend it by adding unseen colors to the test set (TEST-LARGEC) in the format $[c_1, c_2, c_3, c_4]$, where $c_i = 0$ for $i \neq 2$ if $c_2 = 1$ and $c_i \in [0, 1]$ for $i \neq 2$ if $c_2 = 0$, i.e. there is no new colors that have nonzero values in a green channel. This can be interpreted as adding mixtures of red, blue and transparency channels, with nine possible colors in total as opposed to three in the training set (Figure 2).

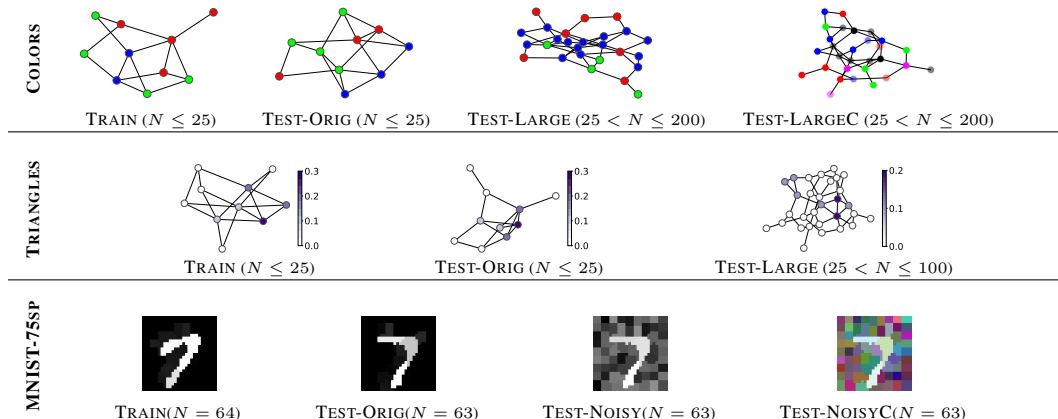

Figure 2: Examples from training and test sets. For COLORS, the correct label is $N_{green} = 4$ in all cases; for TRIANGLES $N_{tri} = 3$ and color intensities denote ground truth attention values $\alpha^{GT}$. The range of the number of nodes, $N$, is shown in each case. For MNIST-75SP, we visualize graphs for digit 7 by assigning an average intensity value to all pixels within a superpixel. Even though superpixels have certain shapes and borders between each other (visible only on noisy graphs), we feed only superpixel intensities and coordinates of their centers of masses to our GNNs.

Neural networks (NNs) have been observed to be brittle if they are fed with test samples corrupted in a subtle way, i.e. by adding a noise [23] or changing a sample in an adversarial way [24], such that a human can still recognize them fairly well. To study this problem, test sets of standard image benchmarks have been enlarged by adding corrupted images [25].

Graph neural networks, as a particular case of NNs, inherit this weakness. The attention mechanism, if designed and trained properly, can improve a net's robustness by attending to only important and ignoring misleading parts (nodes) of data. In this work, we explore the ability of GNNs with and without attention to generalize to noisy graphs and unseen node features. This should help us to understand the limits of GNNs, and potentially NNs in general, with attention and conditions when it succeeds and when it does not. To this end, we generate two additional test sets for MNIST-75SP. In the first set, TEST-NOISY, we add Gaussian noise, drawn from $\mathcal{N}(0, 0.4)$, to superpixel intensity features, i.e. the shape and coordinates of superpixels are the same as in the original clean test set. In the second set, TEST-NOISY-C, we colorize images by adding two more channels and add independent Gaussian noise, drawn from $\mathcal{N}(0, 0.6)$, to each channel (Figure 2).

### 3.3 Network architectures and training

We build 2 layer GNNs for COLORS and 3 layer GNNs for other tasks with 64 filters in each layer, except for MNIST-75SP where we have more filters. Our baselines are GNNs with global sum or max pooling (gpool), DiffPool [10] and top-k pooling [11]. We add two layers of our pooling for TRIANGLES, each of which is a GNN with 3 layers and 32 filters (Eq. 4); whereas a single pooling layer in the form of vector **p** is used in other cases. We train all models with Adam [26], learning rate 1e-3, batch size 32, weight decay 1e-4 (see the *Supp. Material* for details).

For COLORS and TRIANGLES we minimize the regression loss (MSE) and cross entropy (CE) for other tasks, denoted as $\mathcal{L}_{MSE/CE}$. For experiments with supervised and weakly-supervised (described below in Section 3.4) attention, we additionally minimize the Kullback-Leibler (KL) divergence loss between ground truth attention $\alpha^{GT}$ and predicted coefficients $\alpha$. The KL term is weighted by scale $\beta$, so that the total loss for some training graph with $N$ nodes becomes:

$$\mathcal{L} = \mathcal{L}_{MSE/CE} + \frac{\beta}{N} \sum_i \alpha_i^{GT} \log(\frac{\alpha_i^{GT}}{\alpha_i}). \qquad (5)$$

We repeat experiments at least 10 times and report an average accuracy and standard deviation in Tables 1 and 2. For COLORS we run experiments 100 times, since we observe larger variance. In Table 1 we report results on all test subsets independently. In all other experiments on COLORS, TRIANGLES and MNIST-75SP, we report an average accuracy on the combined test set. For COLLAB, PROTEINS and D&D, we run experiments 10 times using splits described in Section 3.1.

The only hyperparameters that we tune in our experiments are threshold $\tilde{\alpha}$ in our method (Eq. 3), ratio $r$ in top-k (Eq. 2) and $\beta$ in Eq. 5. For synthetic datasets, we tune them on a validation set generated in the same way as TEST-ORIG. For MNIST-75SP, we use part of the training set. For COLLAB, PROTEINS and D&D, we tune them using 10-fold cross-validation on the training set.

**Attention correctness.** We evaluate attention correctness using area under the ROC curve (AUC) as an alternative to other methods, such as [27], which can be overoptimistic in some extreme cases, such as when all attention is concentrated in a single node or attention is uniformly spread over all nodes. AUC allows us to evaluate the ranking of $\alpha$ instead of their absolute values. Compared to ranking metrics, such as rank correlation, AUC enables us to directly choose a pooling threshold $\tilde{\alpha}$ from the ROC curve by finding a desired balance between false-positives (pooling unimportant nodes) and false-negatives (dropping important nodes).

To evaluate attention correctness of models with global pooling, we follow the idea from convolutional neural networks [28]. After training a model, we remove node $i \in [1, N]$ and compute an absolute difference from prediction $y$ for the original graph:

$$\alpha_i^{WS} = \frac{|y_i - y|}{\sum_{j=1}^{N} |y_j - y|}, \qquad (6)$$

where $y_i$ is a model's prediction for the graph without node $i$. While this method shows surprisingly high AUC in some tasks, it is not built-in in training and thus does not help to train a better model and only implicitly interprets a model's prediction (Figures 5 and 7). However, these results inspired us to design a weakly-supervised method described below.

### 3.4 Weakly-supervised attention supervision

Although for COLORS, TRIANGLES and MNIST-75SP we can define ground truth attention, so that it does not require manual labeling, in practice it is usually not the case and such annotations are hard to define and expensive, or even unclear how to produce. Based on results in Table 1, supervision of attention is necessary to reveal its power. Therefore, we propose a weakly-supervised approach, agnostic to the choice of a dataset and model, that does not require ground truth attention labels, but can improve a model's ability to generalize. Our approach is based on generating attention coefficients $\alpha_i^{WS}$ (Eq. 6) and using them as labels to train our attention model with the loss defined in Eq 5. We apply this approach to COLORS, TRIANGLES and MNIST-75SP and observe peformance and robustness close to supervised models. We also apply it to COLLAB, PROTEINS and D&D, and in all cases we are able to improve results compared to unsupervised attention.

**Training weakly-supervised models.** Assume we want to train model **A** with "weak-sup" attention on a dataset without ground truth attention. We first need to train model **B** that has the same architecture as **A**, but does not have any attention/pooling between graph convolution layers. So, model **B** has only global pooling. After training **B** with the $\mathcal{L}_{MSE/CE}$ loss, we need to evaluate training graphs on **B** in the same way as during computation of $\alpha^{WS}$ in Eq. 6. In particular, for each training graph $\mathcal{G}$ with $N$ nodes, we first make a prediction $y$ for the entire $\mathcal{G}$. Then, for each $i \in [1, N]$, we remove node $i$ from $\mathcal{G}$, and feed this reduced graph with $N-1$ nodes to model **B** recording the model's prediction $y_i$. We then use Eq. 6 to compute $\alpha^{WS}$ based on $y$ and $y_i$. Now, we can train **A** and use $\alpha^{WS}$ instead of ground truth $\alpha^{GT}$ in Eq. 5 to optimize both *MSE/CE* and *KL* losses.

## 4 Analysis of results

In this work, we aim to better understand attention and generalization in graph neural networks, and, based on our empirical findings, below we provide our analysis for the following questions.

**How powerful is attention over nodes in GNNs?** Our results on the COLORS, TRIANGLES and MNIST-75SP datasets suggest that the main strength of attention over nodes in GNNs is the ability to generalize to more complex or noisy graphs at test time. This ability essentially transforms a model that fails to generalize into a fairly robust one. Indeed, a classification accuracy gap for COLORS-LARGEC between the best model without supervised attention (GIN with global pooling) and a

Table 1: **Results on three tasks for different test subsets.** $\pm$ denotes standard deviation, not shown in case of small values (large values are explained in Section 4). ATTN denotes attention accuracy in terms of AUC and is computed for the combined test set. The best result in each column (ignoring upper bound results) is bolded. ▨ denotes poor results with relatively low accuracy and/or high variance; ▨ denotes failed cases with accuracy close to random and/or extremely high variance. † For COLORS and MNIST-75SP, ChebyNets are used instead of ChebyGINs as described in the *Supp. Material*.

|  |  | **COLORS** | | | | **TRIANGLES** | | | **MNIST-75SP** | | | |
|---|---|---|---|---|---|---|---|---|---|---|---|---|
|  |  | ORIG | LARGE | LARGEC | ATTN | ORIG | LARGE | ATTN | ORIG | NOISY | NOISYC | ATTN |
| Global pool | GCN | 97 | 72±15 | 20±3 | 99.6 | 46±1 | 23±1 | 79 | 78.3±2 | 38±4 | 36±4 | 72±2 |
| | GIN | 96±10 | 71±22 | 26±11 | 99.2 | 50±1 | 22±1 | 77 | 87.6±3 | 55±11 | 51±12 | 71±5 |
| | ChebyGIN† | **100** | 93±12 | 15±7 | 99.8 | 66±1 | 30±1 | 79 | **97.4** | 80±12 | 79±11 | 72±3 |
| Unsuperv. | GIN, top-k | 99.6 | 17±4 | 9±3 | 75±6 | 47±2 | 18±1 | 63±5 | 86±6 | 59±26 | 55±23 | 65±34 |
| | GIN, ours | 94±18 | 13±7 | 11±6 | 72±15 | 47±3 | 20±2 | 68±3 | 82.6±8 | 51±28 | 47±24 | 58±31 |
| | ChebyGIN†, top-k | **100** | 11±7 | 6±6 | 79±20 | 64±5 | 25±2 | 76±6 | 92.9±4 | 68±26 | 67±25 | 52±37 |
| | ChebyGIN†, ours | 80±30 | 16±10 | 11±6 | 67±31 | 67±3 | 26±2 | 77±4 | 94.6±3 | 80±23 | 77±22 | 78±31 |
| Supervised | GIN, topk | 87±1 | 39±18 | 28±8 | **99.9** | 49±1 | 20±1 | 88 | 90.5±1 | 85.5±2 | 79±5 | 99.3 |
| | GIN, ours | **100** | **96±9** | **89±18** | 99.8 | 49±1 | 22±1 | 76±1 | 90.9±0.4 | 85.0±1 | 80±3 | 99.3 |
| | ChebyGIN†, topk | **100** | 86±15 | 31±15 | 99.8 | 83±1 | 39±1 | **97** | 95.1±0.3 | 90.6±0.8 | 83±16 | **100** |
| | ChebyGIN†, ours | **100** | 94±8 | 75±17 | 99.8 | **88±1** | **48±1** | 96 | 95.4±0.2 | **92.3±0.4** | **86±16** | **100** |
| Weak sup. | ChebyGIN†, ours | **100** | 90±6 | 73±14 | **99.9** | 68±1 | 30±1 | 88 | 95.8±0.4 | 88.8±4 | **86±9** | 96.5±1 |
| Upper bound | GIN | 100 | 100 | 100 | 100 | 94±1 | 85±2 | 100 | 93.6±0.4 | 90.8±1 | 90.8±1 | 100 |
| | ChebyGIN† | 100 | 100 | 100 | 100 | 99.8 | 99.4±1 | 100 | 96.9±0.1 | 94.8±0.3 | 95.1±0.3 | 100 |

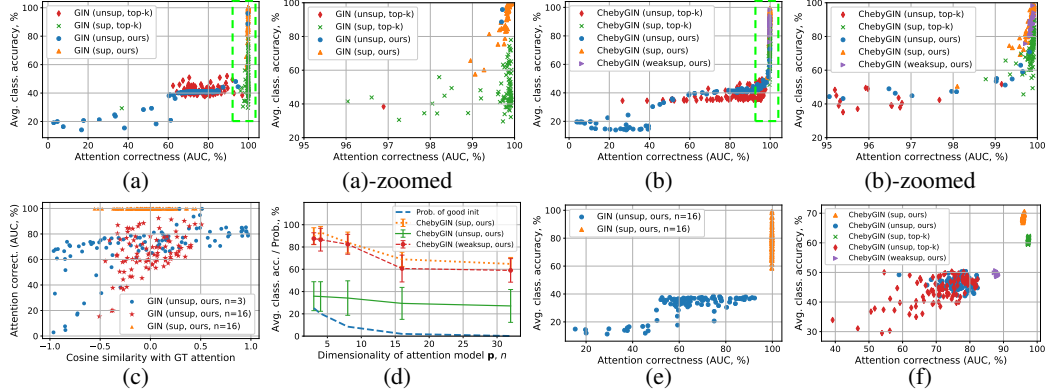

Figure 3: Disentangling factors influencing attention and classification accuracy for COLORS *(a-e)* and TRIANGLES *(f)*. Accuracies are computed over all test subsets. Notice the exponential growth of classification accuracy depending on attention correctness *(a,b)*, see zoomed plots *(a)*-zoomed, *(b)*-zoomed for cases when attention AUC>95%. *(d)* Probability of a good initialization is estimated as the proportion of cases when cosine similarity > 0.5; error bars indicate standard deviation. *(c-e)* show results using a higher dimensional attention model, $\mathbf{p} \in \mathbb{R}^n$.

similar model with supervised attention (GIN, sup) is more than 60%. For TRIANGLES-LARGE this gap is 18% and for MNIST-75SP-NOISY it is more than 12%. This gap is even larger if compared to upper bound cases indicating that our supervised models can be further tuned and improved. Models with supervised or weakly-supervised attention also have a more narrow spread of results (Figure 3).

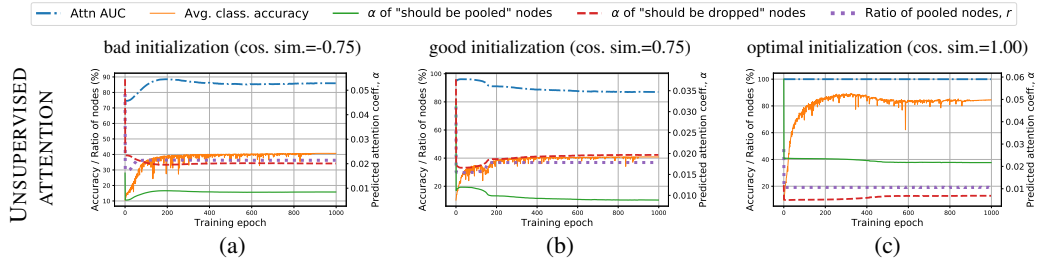

Figure 4: Influence of initialization on training dynamics for COLORS using GIN trained in the unsupervised way. For the supervised cases, see the *Supp. Material*. The nodes that should be pooled according to our ground truth prior, must have larger attention values $\alpha$. However, in the unsupervised cases, only the model with an optimal initialization *(c)* reaches a high accuracy, while other models *(a,b)* are stuck in a suboptimal state and wrong nodes are pooled, which degrades performance. In these experiments, we train models longer to see if they can recover from a bad initialization.

**What are the factors influencing performance of GNNs with attention?** We identify three key factors influencing performance of GNNs with attention: initialization of the attention model (i.e. vector $\mathbf{p}$ or GNN in Eq. 4), strength of the main GNN model (i.e. the model that actually performs classification), and finally other hyperparameters of the attention and GNN models.

We highlight initialization as the critical factor. We ran 100 experiments on COLORS with random initializations (Figure 3, *(a-e)*) of the vector $\mathbf{p}$ and measured how performance of both attention and classification is affected depending on how close (in terms of cosine similarity) the initialized $\mathbf{p}$ was to the optimal one, $\mathbf{p} = [0, 1, 0]$. We disentangle the dependency between the classification accuracy and cos. sim. into two functions to make the relationship clearer (Figure 3, *(a, c)*). Interestingly, we found that classification accuracy depends *exponentially* on attention correctness and becomes close to 100% only when attention is also close to being perfect. In the case of slightly worse attention, even starting from 99%, classification accuracy drops significantly. This is an important finding that can also be valid for other more realistic applications. In the TRIANGLES task we only partially confirm this finding, because our attention models could not achieve AUC high enough to boost classification. However, by observing the upper bound results obtained by training with ground truth attention, we assume that this boost potentially should happen once attention becomes accurate enough.

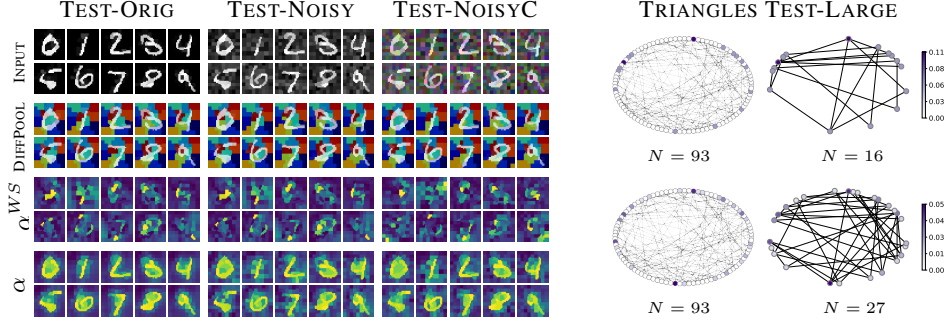

TEST-ORIG   TEST-NOISY   TEST-NOISYC        TRIANGLES TEST-LARGE

Figure 5: Qualitative analysis. For MNIST-75SP (on the left) we show examples of input test images (top row), results of DiffPool [10] (second row), attention weights $\alpha^{WS}$ generated using a model with global pooling based on Eq. 6 (third row), and $\alpha$ predicted by our weakly-supervised model (bottom row). Both our attention-based pooling and DiffPool can be strong and interpretable depending on the task, but in our tasks DiffPool was inferior (see the *Supp. Material*). For TRIANGLES (on the right) we show an example of a test graph with $N = 93$ nodes with six triangles and the results of pooling based on ground truth attention weights $\alpha^{GT}$ (top row); in the bottom row we show attention weights predicted by our weakly-supervised model and results of our threshold-based pooling (Eq. 3). Note that during training, our model has not encountered noisy images (MNIST-75SP) nor graphs larger than with $N = 25$ nodes (TRIANGLES).

**Why is the variance of some results so high?** In Table 1 we report high variance of results, which is mainly due to initialization of the attention model as explained above. This variance is also caused by initialization of other trainable parameters of a GNN, but we show that once the attention model is perfect, other parameters can recover from a bad initialization leading to better results. The opposite, however, is not true: we never observed recovery of a model with poorly initialized attention (Figure 4).

**How top-k compares to our threshold-based pooling method?** Our method to attend and pool nodes (Eq. 3) is based on top-k pooling [11] and we show that the proposed threshold-based pooling is superior in a principle way. When we use supervised attention our results are better by more than 40% on COLORS-LARGEC, by 9% on TRIANGLES-LARGE and by 3% on MNIST-75SP. In Figure 3 (*(a,b)*-zoomed) we show that GIN and ChebyGIN models with supervised top-k pooling never reach an average accuracy of more than 80% as opposed to our method which reaches 100% in many cases.

**How results change with increase of attention model input dimensionality or capacity?** We performed experiments using ChebyGIN-h - a model with higher dimensionality of an input to the attention model (see the *Supp. Material* for details). In such cases, it becomes very unlikely to initialize it in a way close to optimal (Figure 3, *(c-e)*), and attention accuracy is concentrated in the 60-80% region. Effect of the attention model of such low accuracy is neglible or even harmful, especially on the large and noisy graphs. We also experimented with a deeper attention model (ChebyGIN-h), i.e. a 2 layer fully-connected layer with 32 hidden units for COLORS and MNIST-75SP, and a deeper GNN (Eq. 4) for TRIANGLES. This has a positive effect overall, except for TRIANGLES, where our attention models were already deep GNNs.

**Can we improve initialization of attention?** In all our experiments, we initialize $\mathbf{p}$ from the Normal distribution, $\mathcal{N}(0, 1)$. To verify if the performance can be improved by choosing another distribution, we evaluate GIN and GCN models on a wide range of random distributions, Normal $\mathcal{N}(0, \sigma)$ and Uniform $U(-\sigma, \sigma)$, by varying scale $\sigma$ (Figure 6). We found out that for unsupervised training

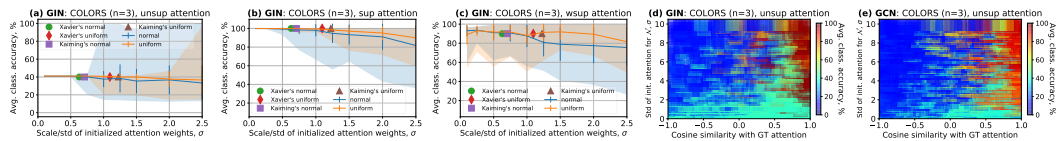

Figure 6: Influence of distribution parameters used to initialize the attention model $\mathbf{p}$ in the COLORS task with $n = 3$ dimensional features. We show points corresponding to the commonly used initialization strategies of Xavier [29] and Kaiming [29]. *(a-c)* Shaded areas show range, bars show $\pm 1$ std. For $n = 16$ see the *Supp. Material*.

(Figure 6, *(a)*), larger initial values and the Normal distribution should be used to make it possible to converge to an optimal solution, which is still unlikely and greatly depends on cosine similarity with GT attention (Figure 6, *(d,e)*). For supervised and "weak-sup" attention, smaller initial weights and either the Normal or Uniform distribution should be used (Figure 6, *(b,c)*).

**What is the recipe for more powerful attention GNNs?** We showed that GNNs with supervised training of attention are significantly more accurate and robust, although in case of a bad initialization it can take a long time to reach the performance of a better initialization. However, supervised attention is often infeasible. We suggested an alternative approach based on weakly-supervised training and validated it on our synthetic (Table 1) and real (Table 2) datasets. In case of COLORS, TRIANGLES and MNIST-75SP we can compare to both unsupervised and supervised models and conclude that our approach shows performance, robustness and relatively low variation (i.e. sensitivity to initialization) similar to supervised models and much better than unsupervised models. In case of COLLAB, PROTEINS and D&D we can only compare to unsupervised and global pooling models and confirm that our method can be effectively employed for a wide diversity of graph classification tasks and attends to more relevant nodes (Figures 5 and 7). Tuning the distribution and scale $\sigma$ for the initialization of attention can further improve results. For instance, on PROTEINS for the weakly-supervised case, we obtain 76.4% as opposed to 76.2%.

Table 2: **Results on the social (COLLAB) and molecule (PROTEINS and D&D) datasets.** We use 3 layer GCNs [18] or ChebyNets [8] (see *Supp. Material* for architecture details). Dataset subscripts denote the maximum number of nodes in the training set according to our splits (Section 3.1).

|  | COLLAB$_{35}$ | PROTEINS$_{25}$ | D&D$_{200}$ | D&D$_{300}$ |
|---|---|---|---|---|
| # train / test graphs | 500 / 4500 | 500 / 613 | 462 / 716 | 500 / 678 |
| # nodes ($N$) train | 32-35 | 4-25 | 30-200 | 30-300 |
| # nodes ($N$) test | 32-492 | 6-620 | 201-5748 | 30-5748 |
| Global max | 65.9±3.4 | 74.4±1.0 | 29.7±4.9 | 72.7±3.6 |
| Unsup, ours | 65.7±3.5 | 75.6±1.4 | 51.9±5.3 | 77.2±2.9 |
| Weak-sup | **67.0±1.7** | **76.2±0.7** | **54.3±5.0** | **78.4±1.1** |

GLOBAL POOL    UNSUP    UNSUP POOLED    WEAK-SUP    WEAK-SUP POOLED

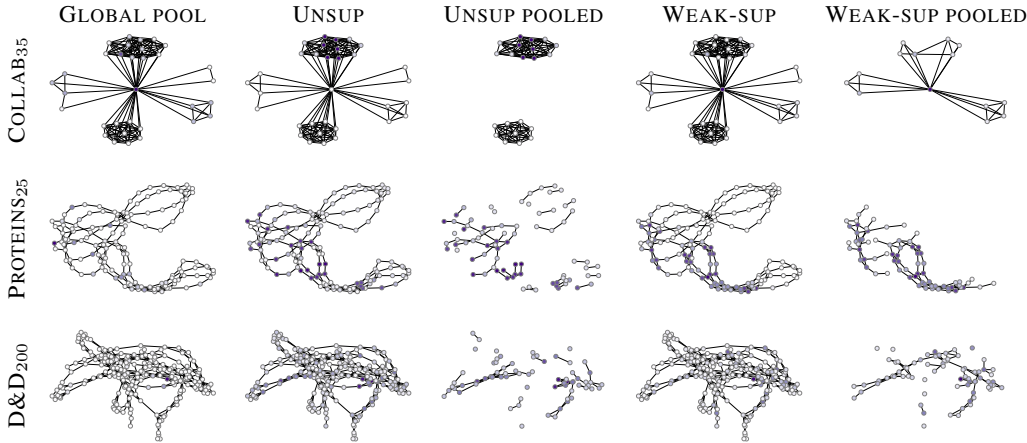

Figure 7: Qualitative results. In COLLAB, a graph represents an ego-network of a researcher, therefore *center nodes* are important. In PROTEINS and D&D, a graph is a protein and nodes are amino acids, so it is important to attend to a *connected chain* of amino acids to distinguish an enzyme from a non-enzyme protein. Our weakly-supervised method attends to and pools more relevant nodes compared to global and unsupervised models, leading to better classification results.

## 5 Conclusion

We have shown that learned attention can be extremely powerful in graph neural networks, but only if it is close to optimal. This is difficult to achieve due to the sensitivity of initialization, especially in the unsupervised setting where we do not have access to ground truth attention. Thus, we have identified initialization of attention models for high dimensional inputs as an important open issue. We also show that attention can make GNNs more robust to larger and noisy graphs, and that the weakly-supervised approach proposed in our work brings advantages similar to the ones of supervised models, yet at the same time can be effectively applied to datasets without annotated attention.

**Acknowledgments**

This research was developed with funding from the Defense Advanced Research Projects Agency (DARPA). The views, opinions and/or findings expressed are those of the author and should not be interpreted as representing the official views or policies of the Department of Defense or the U.S. Government. The authors also acknowledge support from the Canadian Institute for Advanced Research and the Canada Foundation for Innovation. We are also thankful to Angus Galloway for feedback.

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
