[Supplementary Material]

# Supplementary Material to "Understanding Attention and Generalization in Graph Neural Networks"

**Boris Knyazev**
University of Guelph
Vector Institute
bknyazev@uoguelph.ca

**Graham W. Taylor**
University of Guelph
Vector Institute, Canada CIFAR AI Chair
gwtaylor@uoguelph.ca

**Mohamed R. Amer**[*]
Robust.AI
mohamed@robust.ai

## 1.1 Additional results

Figure 1: Influence of initialization on training dynamics for COLORS using GIN trained in the supervised ways. In the supervised cases, models converge to a perfect accuracy and initialization only affects the speed of convergence. In these experiments, we train models longer to see if they can recover from a bad initialization. For the unsupervised cases, see Figure 4.

Figure 2: Influence of distribution parameters used to initialize the attention model $\mathbf{p}$ in the COL-ORS task with $n = 16$ dimensional features and the GIN model. We show points corresponding to the commonly used initialization strategies of (Xavier [1]) and (Kaiming [1]). *(a-c)* Shaded areas show range, bars show $\pm 1$ std. For $n = 3$ see Figure 6. For the GCN model we observe similar trends, but with lower accuracies.

---

[*]Most of this work was done while the author was at SRI International.

Table 1: **Additional results on three tasks for different test subsets.** ChebyGIN-d - deeper attention model. ChebyGIN-h - higher dimensionality of the input fed to the attention model (see Table 2 for architectures).

| | Colors | | | | Triangles | | | MNIST-75sp | | | |
| | Orig | Large | LargeC | Attn | Orig | Large | Attn | Orig | Noisy | NoisyC | Attn |
|---|---|---|---|---|---|---|---|---|---|---|---|
| GIN, global pool | 96±10 | 71±22 | 26±11 | 99.2 | 50±1 | 22±1 | 77 | 87.6±3 | 55±11 | 51±12 | 71±5 |
| GIN, DiffPool [2] | 58±4 | 16±2 | 28±3 | 97 | 39±1 | 18±1 | 82 | 83±1 | 54±6 | 43±3 | 50±2 |
| ChebyGIN-d, unsup, ours | 97±13 | 24±8 | 15±5 | 91±21 | 62±14 | 25±3 | 78±2 | 96.4±1 | 88.4±10 | 88.3±10 | 92±15 |
| ChebyGIN-h, unsup, ours | 67±38 | 15±8 | 1±1 | 69±25 | 59±13 | 25±4 | 76±4 | 95.5±3 | 76±20 | 65±18 | 74±33 |

## 1.2 Additional analysis

**How results differ depending on to which layer we apply the attention model?** When an attention model is attached to deeper layers (as we do for Triangles and MNIST-75sp), the signal that it receives is much stronger compared to the first layers, which positively influences overall performance. But in terms of computational cost, it is desirable to attach an attention model closer to the input layer to reduce graph size in the beginning of a forward pass. Using this strategy is also more reasonable when we know that attention weights can be determined solely by input features (as we do in our Colors task), or when the goal is to interpret model's predictions. In contrast, deeper features contain information about a large neighborhood of nodes, so importance of a particular node represents the importance of an entire neighborhood making attention less interpretable.

**Why is initialization of attention important?** One of the reasons that initialization is so important is because training GNNs with attention is a *chicken or the egg* sort of problem. In order to attend to important nodes, the model needs to have a clear understanding of the graph. Yet, in order to gain that level of understanding, the model needs strong attention to avoid focusing on noisy nodes. During training, the attention model predicts attention coefficients $\alpha$ and they might be wrong, especially at the beginning of training, but the rest of the GNN model assumes those predictions to be correct and updates its parameters according to those $\alpha$. This problem is revealed by taking the gradient of an attention function (Eq. 1): $Z = \alpha \odot X$, where $X = f(w, \cdot)$ are node features, and $f$ is some differentiable function with parameters $w$ used to propagate node features: $\frac{\partial Z}{\partial w} = \frac{\partial Z}{\partial f}\frac{\partial f}{\partial w} = \alpha\frac{\partial f}{\partial w}$. Gradients $\frac{\partial Z}{\partial w}$, that are used to update parameters $w$ in gradient descent, reinforce potentially wrong predictions $\alpha$, since they depend on $\alpha$, and the model solution can diverge from the optimal one, which we observe in Figure 4 *(a,b)*. Hence, the performance of such a model largely depends on the initial state, i.e. how accurate were $\alpha$ after the first forward pass.

## 1.3 Dataset statistics and model hyperparameters

Table 2: **Dataset statistics and model hyperparameters for our controlled environment experiments.** Hyperparameters $\tilde{\alpha}$ and $r$ are chosen based on the validation sets.

| | COLORS | TRIANGLES | MNIST-75SP |
|---|---|---|---|
| # train graphs | 500 | 30,000 | 60,000 |
| # val graphs | 2,500 | 5,000 | 5,000 (from the training set) |
| # test graphs ORIG | 2,500 | 5,000 | 10,000 |
| # test graphs LARGE/NOISY | 2,500 | 5,000 | 10,000 |
| # test graphs LARGEC/NOISYC | 2,500 | — | 10,000 |
| # classes | 11 | 10 | 10 |
| # nodes ($N$) train/val | 4-25 | 4-25 | <=75 |
| # nodes ($N$) test | 4-200 | 4-100 | <=75 |
| # layers and filters | 2 layers, 64 filters in each | 3 layers, 64 filters in each | 3 layers: 4, 64, 512 filters |
| Dropout | 0 | 0 | 0.5 |
| Nonlinearity | ReLU | ReLU | ReLU |
| # pooling layers | 1 | 2 | 1 |
| READOUT layer | global sum | global max | global max |
| GIN aggregator | SUM 2 layer MLP with 256 hid. units | SUM 2 layer MLP with 64 hid. units | SUM 2 layer MLP with 64 hid. units |
| ChebyGIN aggregator | MEAN 1 layer MLP[3] | SUM 2 layer MLP with 64 hid. units | MEAN 1 layer MLP[3] |
| ChebyGIN max scale, $K$ | 2 | 7 | 4 |
| Attention model | **p** applied to input layer[4] | Same arch. as the class. GNN, but $K = 2$ for ChebyGIN, applied to hidden layer (Eq. 4) | **p** applied to hidden layer[5] |
| Default initialization | $\mathcal{N}(0, 1)$ | $U(-a, a)$ for linear layers according to [1] in PyTorch | $\mathcal{N}(0, 1)$ |
| Optimal weights of attention model | collinear to $\mathbf{p} = [0, 1, 0]$ | Unknown | Unknown |
| Ground truth attention for node $i$ | $\alpha_i^{GT} = 1/N_{green}$ | $\alpha_i^{GT} = T_i / \sum_i T_i$, $T_i$ is the number of triangles that include node $i$ | $\alpha_i^{GT} = 1/N_{nonzero}$, where $i$ - indices of super-pixels (nodes) with nonzero intensity, $N_{nonzero}$ - total number of such superpixels; $\alpha_i^{GT} = 0$ for other nodes[6] |
| Attention model of ChebyGIN-d | 2 layer MLP with 32 hid. units | 4 layer GNN with 32 filters | 2 layer MLP with 32 hid. units |
| Attention model of ChebyGIN-h | 32 features in the input instead of 4 | 128 filters in the first layer instead of 64 | 32 filters in the first layer instead of 4 |
| Optimal threshold, $\tilde{\alpha}$ | chosen in the range from 0.0001 to 0.1 (usually values around $1/N$ are the best) | | |
| Example of used $\tilde{\alpha}$ | 0.03 – unsup, 0.05 – sup | 0.0001 – unsup, 0.001 – sup, 0.01 – weak-sup | 0.01 |
| Optimal ratio, $r$ | chosen in the range from 0.05 to 1.0 with step 0.02-0.05 (usually values close to 1.0 are the best) | | |
| Example of used $r$ | 1.0 | 1.0 – unsup, 0.97 – sup | 0.3 |
| $\beta$ in loss (Eq. 5 in the paper) | 100 | | |
| Number of clusters in DiffPool | 4[1] | 4[1] | 25 |
| Training params | 100 epochs (lr decay after 90)[2] Models with attn: 300 epochs (lr decay after 280) | 100 epochs (lr decay after 85 and 95 epochs) | 30 epochs (lr decay after 20 and 25 epochs) |

[1]In DiffPool, the number of clusters returned after pooling must be fixed before we start training. While this number can be smaller or larger than the number of nodes in the graph, we still did not find it beneficial to use DiffPool with a number of clusters larger than 4 (the minimal number of nodes in training graphs). Part of the issue is that we train on small graphs and test on large ones and it is hard to choose the number of clusters suitable for graphs of all sizes.

[2]Fewer than for attention models, since they converged faster.

[3]We found that using the SUM aggregator and 2 layer MLPs is not necessary for COLORS and MNIST-75SP, since the tasks are relatively easy and the standard ChebyNet models performed

comparably. For MNIST-75SP, the SUM aggregator and 2 layer MLPs were also unstable during training.

[4] Since perfect attention weights can be predicted solely based on input features.

[5] Attention applied to a hidden layer receives a stronger signal compared when applied to the input layer, which improves results and makes it unnecessary to the use a GNN to predict attention weights as we do for TRIANGLES.

[6] For supervised and weakly-supervised models, we found it useful to set $\alpha_i^{GT} = 0$ for nodes with superpixel intensity smaller than 0.5.

Table 3: **Dataset statistics and model hyperparameters for experiments with unavailable ground truth attention.** Dataset subscripts denote the maximum number of nodes in the training set according to our splits. *In COLLAB nodes do not have any features and a common practice is to add one-hot node degree features, in the same way as we do for TRIANGLES. The range of node degrees is from 0 to 491, hence the input dimensionality is 492. Results are reported after repeating the experiments 100 times: 10 seeds defining train/test splits $\times$ 10 seeds defining model parameters. Hyperparameters $\tilde{\alpha}$ and $\beta$ are chosen based on 10-fold cross-validation on the training sets. Since the training sets are small in these datasets, it is challenging to tune hyperparameters this way. Therefore, in some cases, we adopt a strategy as in [3] and fix hyperparameters for all folds.

| | COLLAB$_{35}$ | PROTEINS$_{25}$ | D&D$_{200}$ | D&D$_{300}$ |
|---|---|---|---|---|
| # input dimensionality | 492* | 3 | 89 | 89 |
| # train graphs | 500 | 500 | 462 | 500 |
| # test graphs | 4500 | 613 | 716 | 678 |
| # classes | 3 (physics research areas) | 2 (enzyme vs non-enzyme) | | |
| # nodes ($N$) train | 32-35 | 4-25 | 30-200 | 30-300 |
| # nodes ($N$) test | 32-492 | 6-620 | 201-5748 | 30-5748 |
| # layers and filters | 3 layers, 64 filters in each, followed by a classification layer | | | |
| Dropout | 0.1 | | | |
| Nonlinearity | ReLU | | | |
| # pooling layers | 1 | | | |
| READOUT layer | global max | | | |
| ChebyGIN aggregator | MEAN, 1 layer MLP (i.e. equivalent to GCN [4] if $K = 1$ or ChebyNet [5] if $K = 3$) | | | |
| ChebyGIN max scale, $K$ | 3 | 1 | 3 | 3 |
| Optimal threshold, $\tilde{\alpha}$ | chosen in the range from 0.0001 to 0.1 | | | |
| Example of used $\tilde{\alpha}$ | 0.002 | 0.0001 for unsup, 0.1 for weak-sup | 0.005 | 0.01 |
| $\beta$ in loss (Eq. 5 in the paper) | chosen in the range from 0.1 to 100 | | | |
| Example of used $\beta$ | 0.5 | 10 | 10 | 0.1 |
| Attention model | 2 layer MLP with 32 hidden units applied to hidden layer | **p** applied to hidden layer | 2 layer MLP with 32 hidden units applied to hidden layer | |
| Default initialization | $U(-a, a)$ for linear layers according to [1] in PyTorch | | | |
| Training params | 50 epochs (lr decay after 25, 35 and 45 epochs) | | | |