[Reviews · NeurIPS 2019]

Reviewer 1



UPDATE: I have increased the score to 6 as long as the authors will revise the paper as promised in the responses. === This paper has more than one topic being discussed. It at the first part talks mostly about the attention mechanism, and in the second section it introduces a new model ChebyGIN, then in the third section it proposed a weakly-supervised attention training approach. Overall, the paper is not all about its title "Understanding Attention in Graph Neural Networks". In 2.3 the paper says "the performance of both GCNs and GINs is quite poor and, consequently, it is also hard for the attention subnetwork to learn", thus it proposes ChebyGIN as a stronger model. In the experiments we only see a few results from non-ChebyGIN models. It raise the concern that are most of the statements and observations on attention only work with stronger models? Or, is ChebyGIN the only strong model that works well with attention? To summarize, it is concerning to use a limited number of models to understand attention. In the section "Why is the variance of some results so high?", the paper raises an interesting issue of attention, which is poorly initialized attention cannot be recovered. It is an important issue as an initialization-sensitive model training is what people would like to avoid. However, there's no further discussion or attempt in solving it. In other deep learning models, for instance CNN, initialization has been studied in several literatures. Different random number distribution may have substantial impact on the initialization. The proposed weakly-supervised attention supervision borrows the attention coefficients from Eq.6, which uses a model's prediction y_i in the computation. The paper is not clear on, or we may have overlooked, how is the model predicting y_i trained? Is it suggesting we first train a model in an unsupervised manner, and use this first version to computer the attention coefficients, then again train the model with the coefficient in the weakly-supervised manner? In Table 2 the variance of the results are still high enough to make the improvement insignificant, especially when comparing unsupervised v.s. weakly-supervised. It is concerning to consider if the proposed weakly-supervised training approach is meaningful without a good initialization.

Reviewer 2



The authors extensively cite the prior work that they are extending here and their work in context. The idea to prune nodes on basis of attention scores they get is novel compared to previous work, as is the case for supervising said attention (atleast in context of graph neural networks). The synthetic datasets provide reasonable test cases to test future algorithms in this domain. I find a few questions that need to be resolved and/or explained in more detail. First, is the attention only used to prune the nodes in the graph or are the representation to the subsequent layers of the model weighted by the representation scores ? If it is the first case, then how is loss function of the final task backproped through the layer i.e if input to next layer is X[X.alpha < threshold], then I don't see a obvious way of passing the gradient through alpha values. If it is the second case, then I would assume the representation passed to subsequent layers are on a reduced scale (and depends on how peaky or uniform the attention is) . In general, authors make a good job of evaluating their design decisions. Some of the examples are - 1) Generalisation performance - In Colors and Triangles task, they test on larger graphs than those appearing in the training set. On Colors and MNIST data, they test on augmenting the feature space to unseen colors. A question I have here why in colors task, GIN performs better while in triangles task , chebyGIN performs better ? Please also clarify how AUROC is being used to quantify attention correctness. Do we consider attention as providing us with binary decision about node's importance i.e if the node is important or not ? If ranking needs to be evaluated, authors might use direct measure of rank measurement like spearman rho or kendall tau. 2) Evaluating GIN and chebyGIN model on all 3 datasets with unsupervised, weakly supervised and supervised attention. The results here show that supervised attention is required for improving performance and that unsupervised attention doesn't give any reasonable improvement of previous baselines. A question I have here is how is weak supervision generated for the model. The authors mention that weak supervision is generated by removing a node from a graph and see how much model's output change . But which model are we talking about here ? Are the authors training a unsupervised attention model and generate weak supervision on basis of that ? In general, Section 3.4 needs to be expanded to detail how weak supervision is generated since in most cases, as author's mention, we don't have ground truth attention and from Table 1, we see that attention only helps if supervised or weakly supervised. 3) Multiple ablation studies on how attention init, thresholding and input dimensionality affect the model output. In section "How results differ depending on to which layer we apply the attention model?" , I am not sure which results/graphs are being used to back this claim (Please clarify that).

Reviewer 3



This paper studies the limitations of attention mechanism in GNNs when conducting graph classification tasks. Through empirical study on graph with ground truth attention, graph with Gaussian noise, and graphs with unseen node features, Overall, it is an interesting work with empirical analysis, while the technical contribution is limited. It provides insights and interesting discussion on the capability of attention mechanism in GNNs; for example, main strength of attention over nodes in GNNs is the ability to generalize to more complex or noisy graphs at test time. Also, the factors influencing performance of GNNs with attention are: initialization of the attention model, strength of the main GNN model, and other hyperparameters of the attention and GNN models. Authors finally suggest that GNNs with supervised training of attention are significantly more accurate and robust, even with weakly-supervised training since ground truth attention is often not available. Their introduced way of supervised or weakly-supervised strategies require important hyperparameters setting, which was not discussed. Most of the results are reported on datasets which have ground truth attention values. Only Table 2 shows evaluation on other real-world datasets, without comparison to GIN and GCN, which had good performance in Table 1.

[Author Response · NeurIPS 2019]

We thank all reviewers (denoted as R1, R2 and R3) for constructive feedback and questions. We provide answers below.

**(R1, R2, R3) Training weakly-supervised models.** Assume we want to train model **A** with "weak-sup" attention
on a dataset w/o ground truth attention. We first need to train model **B** that has the same architecture as **A**, but does not
have any attention/pooling between graph conv. layers. So, model **B** has only global pooling. After training **B** with the
$\mathcal{L}_{MSE/CE}$ loss, we need to evaluate training graphs on **B** as follows. For each training graph $\mathcal{G}$ with $N$ nodes, we first
make a prediction $y$ for the entire $\mathcal{G}$. Then, for each $i \in [1, N]$, we remove node $i$ from $\mathcal{G}$, and feed this reduced graph
with $N-1$ nodes to model **B** and record the model's prediction $y_i$. We then use Eq. 6 to compute $\alpha^{WS}$ based on $y$ and
$y_i$. Now, we can train **A** and use $\alpha^{WS}$ instead of ground truth $\alpha^{GT}$ in Eq. 5 to optimize both *MSE/CE* and *KL* losses.

**(R1) Focus and a title of the paper.** Instead of "*Understanding Attention...*" we propose the new title "*On Initialization*
*and Supervision of Attention...*". The main purpose of incorporating other topics was to support our conclusions about
*attention*, which is our central focus. It can often be relatively easy to identify a phenomenon in (graph) neural networks,
but it is hard to resolve it. Thus, our weak-sup method complements our more analysis rather than methods-driven focus.

**(R1) Limited number of models.** We evaluate and draw conclusions
based on three models of different strength: GIN, ChebyNet, ChebyGIN.
We will also add results of GCN supporting our conclusions (Table 1
and Figure 1). Many other GNNs proposed in the literature are slight
modifications of these. Generally, stronger models are required to achieve
higher attention accuracy, which would lead to the exponential gains
in classification accuracy that we observe. With weaker models, this
phenomenon can be observed on rather simple datasets (COLORS). Note
that in Table 1 of the submitted paper, for COLORS and MNIST-75sp,
ChebyGINs are equivalent to ChebyNets as described in Table 1 of
the Supplementary material and elaborated on following that table (see
footnote 3). We will update Table 1 in the paper to make it clear.

Table 1: **(R1, R3) About Table 2.** In addition to the results of ChebyNet in the submitted paper, we report results of GCN. For more reliable comparison, we repeat experiments for 100 random seeds instead of 10. "init tune" denotes tuning $\sigma$ and choosing between $\mathcal{N}$ or $U$ (see Figure 1 at the bottom); tuning is done in the same way as for other hyperparameters.

| Model | Proteins$_{25}$ |
|---|---|
| GCN + Global max | $74.4_{\pm 1.0}$ |
| GCN + Unsup | $75.6_{\pm 1.4}$ |
| GCN + Weaksup | $76.2_{\pm 0.7}$ |
| GCN + Unsup+init tune | $75.5_{\pm 1.7}$ |
| GCN + Weaksup+init tune | $\mathbf{76.4_{\pm 0.7}}$ |

**(R2) Hard or soft attention scores?** In our model, the features are
*weighted* by attention scores according to Eq. 3, so it is soft. In this
case, the features indeed reduce their scale. But we haven't found this
problematic in our tasks. For really large graphs, we found that using
a constant multiplier $c > 1$ can help considerably to keep the scale of
features in a reasonable range, while still permitting weighting of node
features: $c(\alpha_i X_i)$. This is more an implementation trick and we are releasing code to support such cases. **(R2) Why**
**GIN is better than ChebyGIN in some cases?** ChebyGIN has larger capacity and, thus, can overfit easily to such
a simple training distribution as in COLORS. Moreover, the difference between GIN and ChebyGIN in COLORS is
not significant (see the standard deviations). **(R2) Computing AUC vs Rank correlation.** AUC is computed between
binarized ground truth attention scores (i.e. $\alpha^{GT} > 0$) and predicted $\alpha$. We indeed have considered other metrics.
While they can be good for evaluation, the choice of AUC is more natural, since we can directly choose a pooling
threshold $\tilde{\alpha}$ by looking at the ROC curve and finding a good balance between false-positives (pooling unimportant
nodes) and false-negatives (dropping important nodes). So, AUC provides a comprehensive picture for different $\tilde{\alpha}$.
Also, the problem with rank correlation metrics is that it is unclear whether to include $\alpha$ of dropped nodes during
calculation of the metric, and that can lead to very different results, which complicates comparison.

**(R3) Tuning threshold $\tilde{\alpha}$ and other hyperparams.** The best $\tilde{\alpha}$ is typically around $1/4N$ to $2/N$, where $N$ is the max
number of nodes in the training graphs. We chose $\tilde{\alpha}$ from the range $[0.0001, 0.1]$. We tune $\tilde{\alpha}$ and a few other hyper-
parameters ($r$ in top-k and $\beta$ in Eq. 5) on the val. sets generated in the same way as TEST-ORIG for synthetic datasets;
part of the training set for MNIST-75sp; using 10-fold cross-val. on the training set for COLLAB, PROTEINS and D&D.

Figure 1: **(R1) Initialization methods.** In these experiments, we evaluate **GIN** *(a-d)* and **GCN** *(e)* on a wide range of random distributions $\mathcal{N}(0, \sigma)$ and $U(-\sigma, \sigma)$ by varying $\sigma$. In the submitted version of the paper, we used $\mathcal{N}(0, 1)$ by default. We show points corresponding to the commonly used initialization strategies of (Xavier Glorot & Bengio, 2010) and (Kaiming He et al., 2015). We see that for unsupervised training *(a)*, larger initial values and the Normal distribution should be used to make it possible to converge to an optimal solution, which is still unlikely and greatly depends on cosine similarity with GT attention *(d,e)*. For supervised and "weak-sup" attention, we should use smaller initial weights and either the Normal or Uniform distribution *(b,c)*. We have similar plots for COLORS with $n = 16$ dimensional features to be added to the camera-ready version. *(a-c)* Shaded areas show range, bars show $\pm 1$ std.

[Meta-Review · NeurIPS 2019]

This paper explores node-wise attention in graph neural networks, with the aim of characterizing when it works well. The authors demonstrate that attention often affords only marginal benefits. They propose a weakly supervised regime that tends to improve performance. The experiments are thorough and presented well. Reviewers have highlighted some presentation issues that should be addressed in future versions of the manuscript.